# Point-RFT: Improving Multimodal Reasoning with Visually Grounded Reinforcement Finetuning

**Minheng Ni[1,2] [†], Zhengyuan Yang[3] [†], Linjie Li[3], Chung-Ching Lin[3], Kevin Lin[3], Wangmeng Zuo[2✉], Lijuan Wang[3✉]**

[1]Hong Kong Polytechnic University    [2]Harbin Institute of Technology    [3]Microsoft

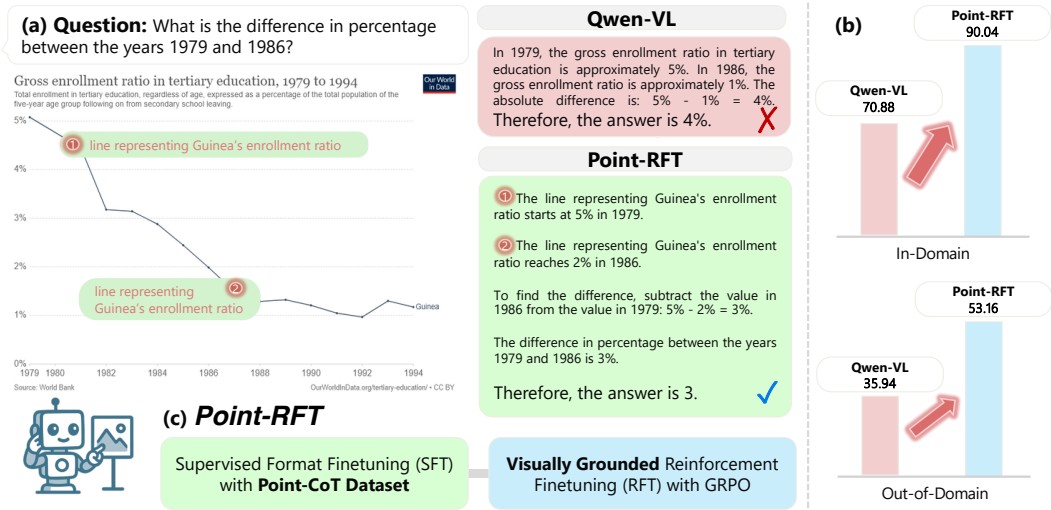

Figure 1: **Point-RFT improves multimodal reinforcement finetuning with visually grounded CoT.** (1) We first construct a Point-CoT dataset for Supervised Format Finetuning (SFT). It allows the model to generate step-by-step reasoning traces explicitly linked to visual pointing, mitigating hallucinations and enhancing perception-reasoning alignment. (2) Reinforcement Finetuning (RFT) with GRPO: Optimizes answer correctness and grounded rationale coherence by rewarding localized visual-textual reasoning paths. Our model, code, and dataset can be found at this link.

## Abstract

Recent advances in large language models have significantly improved textual reasoning through the effective use of Chain-of-Thought (CoT) and reinforcement learning. However, extending these successes to vision-language tasks remains challenging due to inherent limitations in text-only CoT, such as visual hallucinations and insufficient multimodal integration. In this paper, we introduce Point-RFT, a multimodal reasoning framework explicitly designed to leverage visually grounded CoT reasoning for visual document understanding. Our approach consists of two stages: First, we conduct format finetuning using a curated dataset of 71K diverse visual reasoning problems, each annotated with detailed, step-by-step rationales explicitly grounded to corresponding visual elements. Second, we employ reinforcement finetuning targeting visual document understanding. On ChartQA, our approach improves accuracy from 70.88% (format-finetuned baseline) to 90.04%, surpassing the 83.92% accuracy achieved by reinforcement finetuning relying solely on text-based CoT. The result shows that our grounded CoT is more effective for multimodal reasoning compared with the text-only CoT. Moreover, Point-RFT exhibits superior generalization capability across several out-of-domain visual document reasoning benchmarks, including CharXiv, PlotQA, IconQA, TabMWP, *etc.*, and highlights its potential in complex real-world scenarios.

39th Conference on Neural Information Processing Systems (NeurIPS 2025).

# 1 Introduction

Large reasoning models (o1b, 2024; Guo et al., 2025) have shown impressive capabilities in solving complex text-based problems, such as STEM reasoning (Lightman et al., 2023; Rein et al., 2024) and coding tasks (Jain et al., 2024b; Quan et al., 2025). Central to their success is the large language model's ability to improve responses by expanding the textual Chain-of-Thought (CoT) (Wei et al., 2022), also known as scaling inference-time compute (Brown et al., 2024; Snell et al., 2024). Reinforcement learning (Mnih et al., 2015, 2016; Schulman et al., 2017; Shao et al., 2024) further optimizes these models by maximizing the rewards associated with these generated longer, more comprehensive, and accurate responses. Despite these successes in textual reasoning, adapting them to vision-language tasks remains challenging, with unique open challenges such as imperfect visual perception (Fu et al., 2024) as well as integrating reasoning across both visual and text modalities (Hao et al., 2025).

Inspired by the introduction of visual-centric tokens in vision-language models (Yang et al., 2022a; Jain et al., 2024a; Dong et al., 2023; Yang et al., 2022b; Wang et al., 2022b) to improve visual perception, we present Point-RFT, with the insight that the effective CoT for multimodal reasoning should contain grounded visual tokens. To achieve this, we first perform format finetuning across diverse problem sets to enable Point-RFT to generate grounded CoT rationales. Subsequently, the format-finetuned model undergoes reinforcement finetuning with outcome reward on specialized visual document datasets, allowing it to explore the effective usage of grounded CoT in solving visual document questions.

To curate our format finetuning dataset, we establish a pipeline that integrates step-by-step text rationales generated by a large multimodal model (Hurst et al., 2024) with a grounding model (Deitke et al., 2024) that links relevant reasoning steps explicitly to visual points. The resulting dataset comprises 71K images covering a wide range of question types (Guo et al., 2024). This diversity in the dataset design ensures that the model generalizes its learned formatting strategies broadly, allowing it to explore effective solutions for various problems during subsequent reinforcement finetuning. After the format finetuning for grounded CoT, we conducted reinforcement finetuning (Shao et al., 2024) specifically on the ChartQA dataset (Masry et al., 2022). Experimental results demonstrate notable performance improvements: accuracy increased from 87.21% to 90.04% compared to the Supervised finetuning (SFT) baseline and from 83.92% to 90.04% compared to reinforcement learning using text-only CoT. These results show the effectiveness of our Point-RFT and the grounded CoT approach for enhancing multimodal reasoning capabilities.

Beyond the performance improvements on ChartQA, we note that Point-RFT demonstrates several compelling properties for multimodal reasoning: **(1)** Improved generalization capabilities compared to supervised finetuning and reinforcement learning with text-only CoT, with superior performance on various out-of-domain tasks including ChartXiv (Wang et al., 2024), PlotQA (Methani et al., 2020), IconQA (Lu et al., 2021), TabMWP (Lu et al., 2022b), *etc*. **(2)** Enhanced pointing accuracy not only contributes directly to higher-quality final answers but also provides increased interpretability. This capability allows the disentanglement of perception errors from reasoning errors, facilitating more precise diagnostics and targeted improvements.

Our contributions are summarized as follows.

- We present, Point-RFT, the first multimodal reasoning framework explicitly designed for visually grounded reinforcement fine-tuning. We empirically show that "visually grounded CoT" is more effective for multimodal reasoning compared with text-only thoughts.

- We curate a 71K-example dataset where every reasoning step is aligned with point-level visual references, enabling supervised format finetuning that teaches the model to "think while pointing." We then perform RFT with grounded CoT.

- On ChartQA, Point-RFT boosts accuracy from 70.88 % to 90.04 %. It also achieves the best average score over five out-of-domain benchmarks, demonstrating superior generalization ability and interpretability.

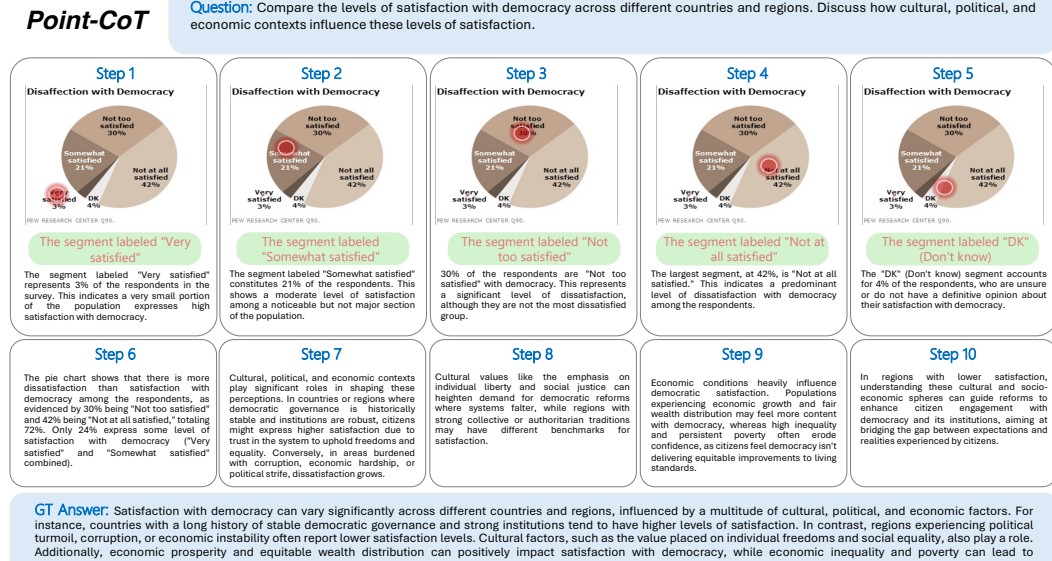

Figure 2: **Visualization of Point-CoT dataset.** Point-CoT dataset integrates the reasoning process of answering questions with point grounding, creating a novel form of multimodal CoT.

## 2 Related Work

**Large Reasoning Language Models.** Recent studies (o1b, 2024; Guo et al., 2025) have demonstrated that reinforcement learning (Mnih et al., 2015, 2016; Schulman et al., 2017; Shao et al., 2024) can significantly enhance the reasoning capabilities of language models (Lightman et al., 2023; Rein et al., 2024; Jain et al., 2024b; Quan et al., 2025). The approach involves the use of chain-of-thought (CoT) reasoning to systematically generate high-quality responses, coupled with outcome-based reward optimization, enabling models to explore and identify effective solutions. However, a purely textual CoT may be sub-optimal for vision-centric (Fu et al., 2024; Rahmanzadehgervi et al., 2024; Hua et al., 2024; Tong et al., 2024) and vision-language tasks (Yu et al., 2023, 2024b; Hao et al., 2025). In this work, we propose and investigate a grounded CoT approach to improve multimodal reasoning.

**Vision-Language Models.** Advancements in vision-language modeling (VLM) (Radford et al., 2021; Yuan et al., 2021; Wang et al., 2022c,a; Yu et al., 2022; Li et al., 2024; Hu et al., 2022) have significantly enhanced the joint understanding of visual and textual inputs, benefiting various downstream tasks such as image captioning (Chen et al., 2015) and visual question answering (Goyal et al., 2017). Recently, contemporary VLMs (Alayrac et al., 2022; OpenAI, 2023a; Yang et al., 2023; Liu et al., 2023; Wang et al., 2023; Chen et al., 2024; Bai et al., 2023; Team, 2023) have integrated multimodal modeling techniques with large language models (LLMs) (OpenAI, 2023b; Grattafiori et al., 2024; Yang et al., 2024), resulting in enhanced capabilities including instruction-following, in-context learning, and improved zero-shot generalization. Nevertheless, the application of reinforcement learning to enhance multimodal reasoning (Xiyao et al., 2024; Zhou et al., 2024; Yu et al., 2024a; Sun et al., 2023; Wang et al., 2025a) remains relatively underexplored due to the unique challenges associated with extending outcome reward-based reinforcement learning approaches from purely textual to multimodal.

**Multimodal Chain-of-Thought.** Multimodal chain-of-thought reasoning reasoning Wang et al. (2025b) has recently attracted intense attention. Early works focus on text-only thoughts for multimodal problems Lu et al. (2022a); Zhang et al. (2024); Xu et al. (2024), showing that explicit step-by-step reasoning boosts visual-question-answering accuracy. More recent studies extend this idea to true multimodal thoughts Rose et al. (2023); Wu et al. (2024); Hu et al. (2024); Fu et al. (2025); Ma et al. (2024), including having models directly generating visualizations, as well as calling vision tools that generate crops and other augmented views. Despite this progress, all existing multimodal chain-of-thought approaches remain predominantly prompt-driven, rather than end-to-end RFT, as explored in this study.

# 3 Method

Our core hypothesis is that reinforcement finetuning with grounded chain-of-thought (CoT) reasoning enhances visual-language reasoning capabilities. To investigate this, we propose a two-stage training framework: (1) supervised fine-tuning (SFT) with format-finetuning dataset, to support grounded CoT, followed by (2) GRPO-based RL optimization with outcome reward signals. Section 3.1 introduces the format fintuning data collection pipeline, and Section 3.2 details the model training approach.

Table 1: **Statistics of dataset.** The final dataset comprises 71K images with diverse question type. This mixture ensures coverage of both free-form and structured visual reasoning scenarios.

| Source | Samples | Ratio | Turn | Points |
|---|---|---|---|---|
| ChartQA | 29533 | 41.5% | 3.97 | 2.02 |
| DVQA | 25269 | 35.5% | 8.78 | 5.79 |
| PlotQA | 14055 | 19.8% | 8.71 | 5.89 |
| CLEVR (MathV360K) | 1322 | 1.9% | 4.61 | 3.43 |
| TallyQA | 931 | 1.3% | 2.93 | 2.01 |

## 3.1 Point-CoT: Format-Finetune Dataset

To enable models to explicitly understand and interact with content in images during multimodal reasoning, we propose the Point-CoT dataset. As shown in Figure 2, the Point-CoT dataset integrates the reasoning process of answering questions with point grounding, creating a novel form of multimodal chain-of-thought (CoT).

Our data generation pipeline addresses two critical challenges in visual-language reasoning: 1) synthesizing logically coherent reasoning chains with explicit visual grounding and 2) ensuring consistency between textual rationales and spatial referring. We design a hybrid pipeline that combines GPT-4o and Molmo-7B, leveraging their complementary strengths: GPT-4o excels at generating human-like reasoning patterns, while Molmo-7B specializes in fine-grained visual localization due to its pretraining on large-scaled geometric annotation tasks. Figure 3 illustrates the overall dataset generation pipeline.

**Step 1: CoT Generation with GPT-4o.** For each image-question-answer triplet from Mammoth-VL datasets, GPT-4o generates comprehensive, step-by-step reasoning traces with text tokens only. Each step may explicitly reference visual elements (*e.g.*, chart components) crucial for deriving the correct and convincing answer. For the referenced visual elements, the model will be required to generate their descriptions, *i.e.*, `<point>...</point>` and the number of objects, `<point_result>...</point_result>`.

**Step 2: Points Grounding Generation with Molmo-7B.** We then use Molmo-7B to annotate coordinates of points $\{x_0, y_0, x_1, y_1, x_2, y_2, ...\}$ for visual elements mentioned in each reasoning step. Coordinates follow raw image resolution rather than normalized scales, and points are formatted as `<points x0="..." y0="..." ...>description</points>` preceding each reasoning step. After point generation, cross-validation on counts of points between GPT-4o rationales, *i.e.*, number in `<point_result>...</point_result>` and Molmo-7B annotations will be conducted, and we will discard points with mismatches after eight retries.

The validation process improves the integrity and accuracy of the data, enhancing the quality of the Chain-of-Thought (CoT) reasoning from two perspectives. First, by using Molmo-7B for visual grounding, hallucinated elements in GPT-4o's reasoning are filtered out because they will fail the grounding process. Second, if Molmo-7B fails to ground or provides incorrect annotations, the number of visual elements mentioned in GPT-4o's text (*e.g.*, "three bars in the chart") will not match the number of instances grounded by Molmo-7B. Thus, if a model's visual perception is accurate, the resulting CoT reasoning will also be accurate. This rigorous process reduces annotation errors compared to single-model approaches.

The final dataset comprises 71K images with diverse question types (counting, comparison, arithmetic) from the subset of Mammoth-VL. This mixture ensures coverage of both free-form and structured visual reasoning scenarios.

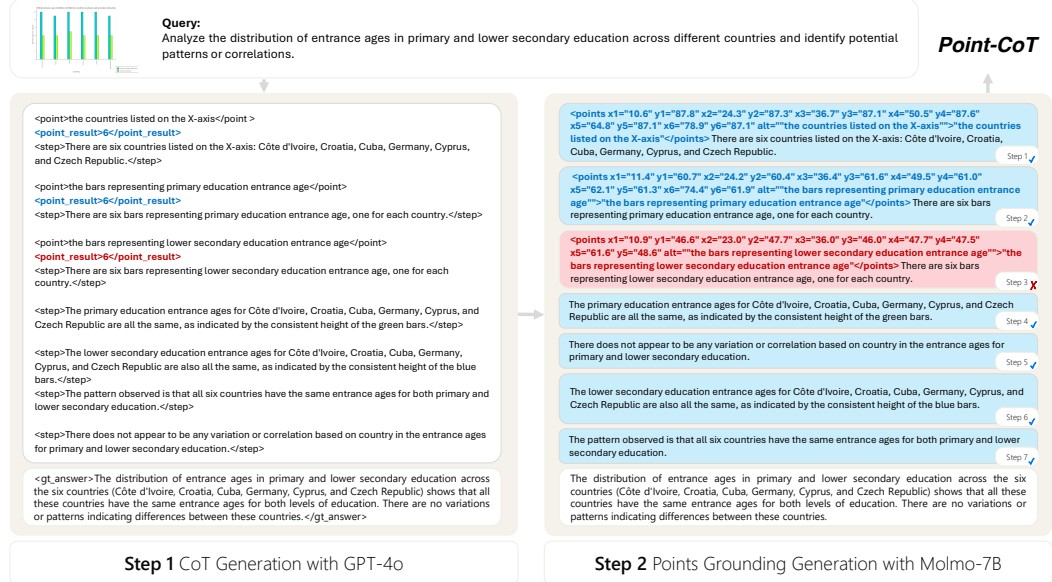

Figure 3: **Overall dataset generation pipeline.** The whole construction process combining LLM reasoning (GPT-4o) and geometric grounding (Molmo-7B). The pipeline ensures spatial-textual consistency through cross-validation, producing our Point-CoT dataset.

## 3.2 Model Training

**Supervised Format Finetuning (SFT) with Point-CoT.** We fine-tune a Qwen2.5-VL as the base model using our format finetuning dataset. Each training instance adheres to the following format:

```
<think>
<points ...>...</points>...
<points ...>...</points>...
...
</think>
<answer>...</answer>
```

The template design serves three purposes: 1) explicitly separating visual grounding from textual reasoning, 2) enabling deterministic parsing of coordinate references during evaluation, and 3) forcing the model to attend to spatial features before generating answers. Our ablation studies indicate that interleaving point annotations with textual content significantly enhance answer accuracy, suggesting that the model benefits from the alignment between spatial referring and thinking in grounded CoT.

Let $\theta$ be model's parameter and $T$ be the length of the text. The loss function is:

$$\mathcal{L}(\theta) := -\mathbb{E}_{(x,y)\sim\mathcal{D}_{\text{Point-CoT}}} \sum_{t=1}^{T} \log P(y_t|x, y_{<t}; \theta), \tag{1}$$

where $(x, y)$ is the query and target response with thinking in dataset $\mathcal{D}_{\text{Point-CoT}}$.

**Reinforcement Finetuning (RFT) with GRPO.** Despite obtaining point-grounded reasoning capability, initial SFT may cause performance drops in OOD domains, suggesting overfitting to synthetic reasoning patterns. To address this, we apply Reinforcement Learning (RL) to further optimize it.

We use the Group-wise Relative Policy Optimization (GRPO) algorithm (Shao et al., 2024) as the optimization method, which extends PPO by introducing task-specific advantage normalization. We implement GRPO with dual rewards to tackle the tension between structural and semantic correctness:

- **Format Reward** ($R_f$) measures structural adherence to the `<think> ... </think>` `<answer> ... </answer>` template via regex matching. It employs the grammar-level regex parser that checks 1) thinking process integrity and 2) whether the answer can be extracted from the answer tag.

- **Accuracy Reward** ($R_a$) computes answer correctness against ground-truth answer using

$$R_a := \mathbb{I}(\hat{y} = y). \tag{2}$$

The combined reward $\hat{R} := R_f + R_a$ is optimized on 15K ChartQA vanilla training data by:

$$\mathcal{J}(\theta) := \mathbb{E}_{(x,y)\sim\mathcal{D}_{\text{ChartQA}}} \frac{\pi_\theta}{\pi_{\theta'}}\hat{R} - \beta \cdot D_{\text{KL}}(\pi_\theta \| \pi_{\text{SFT}}), \tag{3}$$

where $\beta$ is a hyper-parameter of KL divergence, and $\pi_{\text{SFT}}$, $\pi_\theta$, and $\pi_{\theta'}$ is the model after SFT, the optimized model and the model before GRPO optimization.

## 4 Experiment

### 4.1 Implementation Details

All models use AdamW optimizer with $5 \times 10^{-5}$ learning rate, and 512 batch size for SFT and 56 batch size for RL. Training converges in 500 steps for SFT and 100 RL steps with $\beta = 0.00$. We use soft matching for numeric answers (tolerating ±5% relative error) and exact matching for other responses in GRPO. We implement our two-stage training pipeline using PyTorch with $8\times$A100 GPUs based on Easy-R1 (Zheng et al., 2025).

### 4.2 Experimental Setup

We evaluate our approach on six multimodal reasoning benchmark datasets spanning diverse domains:

- **ChartQA** (Masry et al., 2022): A chart understanding task requiring extraction and reasoning over chart elements. We use the official test split for evaluation.
- **CharXiv** (Wang et al., 2024): Scientific charts from academic papers with complex visual encodings. We use the official validation split for evaluation.
- **PlotQA** (Methani et al., 2020): Statistical plots testing trend analysis and numerical reasoning. We randomly sampled 2,000 examples as the test split.
- **IconQA** (Lu et al., 2021): Iconographic reasoning requiring symbolic interpretation. We randomly sampled 2,000 examples as the test split.
- **TabMWP** (Lu et al., 2022b): A task involving tabular math problems that combine table parsing and arithmetic reasoning. We use the official 1,000-sample mini-test split.
- **Counting** (Li et al., 2023): Based on the SuperCLEVR dataset, we sampled 200 examples for evaluation.

We compare the following approaches: (1) **Base-RFT**: A baseline model trained with supervised finetuning on Point-CoT without explicit point annotations, followed by reinforcement learning finetuning with GRPO on ChartQA. This serves as our primary baseline. (2) **Point-SFT**: A model trained with supervised finetuning on Point-CoT with explicit point annotations. (3) **Point-RFT**: A model based on Point-SFT that undergoes reinforcement learning finetuning with GRPO on ChartQA. (4) **Base**: The backbone model without any finetuning is also included in the tables as a reference.

For the base model and Base-RFT, we only require the model to output the reasoning process and the final answer. For Point-SFT and Point-RFT, we additionally require the model to output a reasoning process with explicit point annotations, along with the final answer.

To evaluate all models, we extract the content from the answer tag in their output and compare it to the ground truth to calculate the answer accuracy (Overall) as the primary evaluation metric. In ablation studies, we further analyze the format compliance (Format) and the accuracy of answers within compliant formats (Inner). Note that all models are evaluated with CoT.

### 4.3 Quantitative Results

#### 4.3.1 Main Results

Table 3 presents our main findings across in-domain and out-of-domain test sets. Our Point-RFT achieves state-of-the-art performance on the primary ChartQA benchmark, significantly outperform-

Table 2: **Overall results among different datasets.** Comparison of different training variants on in-domain (ChartQA) and out-of-domain test sets. Point-RFT achieves state-of-the-art in-domain performance (90.04%) and demonstrates strongest OOD generalization (53.16% average), significantly outperforming baselines. The improvement is statistically significant with $p < 0.01$ under $t$-test.

| Method | Backbone | Setting | | In-Domain | Out-of-Domain | | | | | |
| | | Point | RL | ChartQA | CharXiv | PlotQA | IconQA | TabMWP | Counting | Avg. |
|---|---|---|---|---|---|---|---|---|---|---|
| Base | Qwen2.5-VL-7B | - | - | 70.88 | 26.50 | 17.80 | 53.40 | 61.00 | 21.00 | 35.94 |
| Base-RFT | | - | ✓ | 83.92 | 25.20 | 14.85 | 48.55 | 61.40 | 19.00 | 33.80 |
| Point-SFT | Qwen2.5-VL-7B | ✓ | - | 87.21 | 28.00 | 20.34 | 55.25 | 60.30 | 70.00 | 46.78 |
| Point-RFT | | ✓ | ✓ | **90.04** | **36.20** | **20.40** | **59.80** | **70.90** | **78.50** | **53.16** |

Table 3: **Overall results among different models.** The improvement is statistically significant with $p < 0.01$ under $t$-test.

| Method | Backbone | Setting | | In-Domain | Out-of-Domain | | | | | |
| | | CoT | Point | ChartQA | CharXiv | PlotQA | IconQA | TabMWP | Counting | Avg. |
|---|---|---|---|---|---|---|---|---|---|---|
| Base | Qwen2-VL-7B | ✓ | - | 55.40 | 25.20 | 10.50 | **49.15** | 61.00 | 68.00 | 42.77 |
| Point-RFT | | ✓ | ✓ | **86.16** | **28.30** | **18.80** | 46.90 | **61.60** | **68.50** | **44.82** |
| Base | Qwen2.5-VL-3B | ✓ | - | 79.28 | 27.50 | 10.80 | 44.75 | 61.00 | 40.50 | 36.91 |
| Point-RFT | | ✓ | ✓ | **86.24** | **28.80** | **17.90** | **51.00** | **62.20** | **60.50** | **44.08** |
| Base | Qwen2.5-VL-7B | ✓ | - | 70.88 | 26.50 | 17.80 | 53.40 | 61.00 | 21.00 | 35.94 |
| Point-RFT | | ✓ | ✓ | **90.04** | **36.20** | **20.40** | **59.80** | **70.90** | **78.50** | **53.16** |

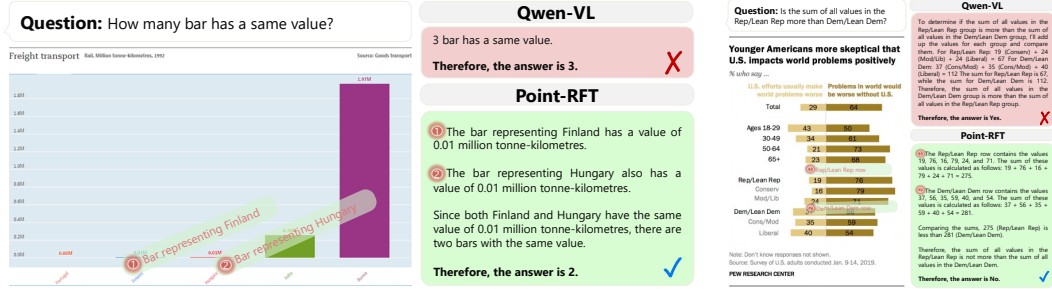

Figure 4: **Cases of In-domain Chart.** These cases highlight the limitations of pure text reasoning in analyzing complex visual elements. Point-RFT successfully reasons by integrating the visual content.

ing the Point-SFT model and text-only reinforcement learning variants, Base-RFT. More importantly, we observe substantial improvements in out-of-domain generalization.

**In-Domain Performance.** On ChartQA, Point-RFT achieves an accuracy of 90.04%, showing a significant improvement over Point-SFT (87.21%), which demonstrates the importance of RL in helping the model learn point-based reasoning. Compared with the text-only reinforcement learning variant (83.92%), the absolute improvement validates that visual grounding significantly enhances reasoning capability through better perception-reasoning alignment. Our method also outperforms the base instruction-tuned Qwen model (70.88%). Since the base model does not perform CoT reasoning or visual localization, Point-RFT can learn advanced visual reasoning patterns.

**Out-of-Domain Generalization.** Point-RFT achieves an average accuracy of 53.16% across five out-of-domain benchmarks, surpassing the text-only reinforcement learning variant by 19.36% absolute. Notably, the gains on PlotQA (+6.55%) and IconQA (+11.25%) suggest that explicit visual references facilitate the transfer of reasoning patterns to novel visual layouts. Furthermore, similar to the in-domain results, Point-RFT significantly outperforms Point-SFT, demonstrating the critical role of RL in acquiring this unique capability. Additionally, we observe that the OOD performance is comparable to the backbone model that undergoes large-scale multi-task instruction finetuning, further validating the generalization capability of Point-RFT.

For extra comparisons, please refer to the **Supplementary Material**.

### 4.3.2 Ablation Studies

**Pointing Format.** Table 4 compares the XML and JSON syntax formats and the impact of point indexing schemes on performance. XML syntax achieves an accuracy of 56.72%, while JSON

Table 4: **Ablation studies of pointing format.** Different coordinate formats and indexing schemes highly influence performance.

| Method | Type | | ChartQA | | |
|---|---|---|---|---|---|
| | Format | Point | Overall | Inner | Format |
| Base | - | - | 79.28 | 82.17 | 96.48 |
| Point-SFT | JSON | ✓ | 49.36 | 71.88 | 69.56 |
| | XML | Random | 31.32 | 75.82 | 39.72 |
| | XML | ✓ | **56.72** | **76.20** | **74.44** |

Table 5: **Ablation studies of point grounding.** Removing visual grounding reduces performance.

| Method | Training | | | ChartQA | | |
|---|---|---|---|---|---|---|
| | Point | SFT | RL | Overall | Inner | Format |
| Base | - | - | - | 79.28 | 82.17 | 96.48 |
| Base-SFT | - | ✓ | - | 27.60 | 75.82 | 36.40 |
| Point-SFT | ✓ | ✓ | - | **56.72** | **76.20** | **74.44** |
| Base-RFT | - | ✓ | ✓ | 77.16 | 79.61 | 96.92 |
| Point-RFT | ✓ | ✓ | ✓ | **86.24** | **86.52** | **99.68** |

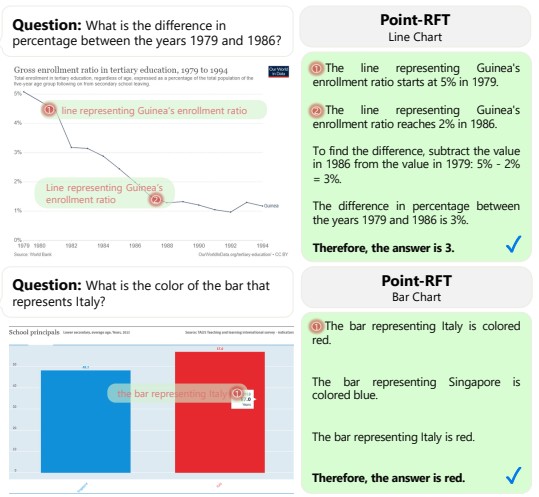

Figure 5: **Case of OOD Chart.** Point-RFT successfully transfers coordinate referencing skills learned from bar charts.

Figure 6: **Cases of different RL steps.** As the steps increased, accurate localization led to correct answers.

achieves 49.36%. This suggests that for more complex reasoning, the XML format better aids the model in generating accurate outputs. Surprisingly, the indexing scheme (0-based vs. 1-based) shows a significant difference (56.72% vs. 31.32%). Interestingly, Qwen's default learned indexing starts from 1. We hypothesize that introducing appropriate variation helps the model better learn the ability to perform pointing during reasoning.

**Explicit Grounding.** We find that removing visual pointing (Table 5) results in severe performance degradation: a 28.5% absolute drop in accuracy for SFT (from 56.72% to 27.60%) and a 3.88% drop for RL (from 81.04% to 77.16%). This validates our hypothesis that explicit visual references are crucial for both initial format learning and subsequent reinforcement training. The 98.32% format compliance rate indicates that structured pointing stabilizes RL optimization, reducing hallucinated reasoning steps. This is critical for real-world applications, where predictable output formats facilitate downstream integration.

For more ablation studies, please refer to the **Supplementary Material**.

## 4.4 Qualitative Results

### 4.4.1 Chart Reasoning

**In-domain Chart** To investigate how visually grounded reasoning impacts the model's inference, we selected two groups of cases to observe the differences between our reasoning and our baseline model Base-RFT in Figure 4. We observe that our model successfully acquires the ability to perform step-by-step reasoning and locate corresponding visual elements, enabling Point-RFT to arrive at the correct answers in these two high-information-density examples. This demonstrates the strengths of Point-RFT, particularly its systematic exploration of visual evidence.

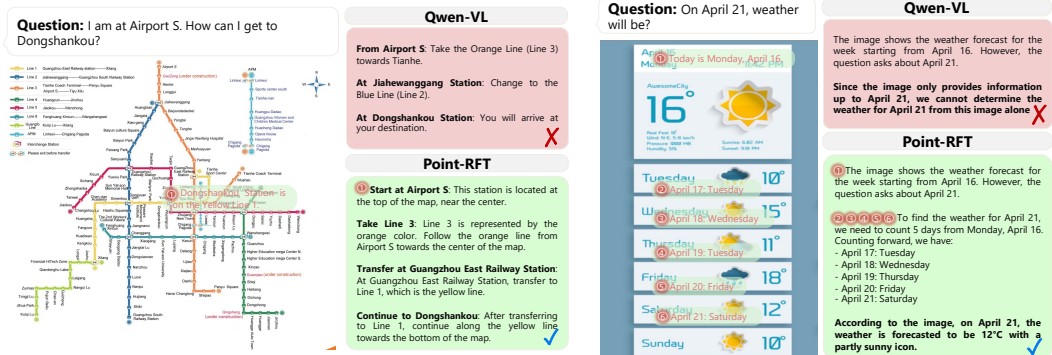

Figure 7: **Cases of generalization.** Point-RFT can perform reasoning on more complex graphics, showcasing its generalization capability.

**Out-of-domain Chart** Figure 5 illustrates how CoT-based explicit grounding facilitates OOD adaptation. When faced with chart types lacking explicit visual elements (*e.g.*, line charts), Point-RFT successfully transfers coordinate referencing skills learned from bar charts. Despite consistent descriptions of visual elements across steps, the model correctly identified different locations. Meanwhile, the model was able to accurately locate bars containing special shapes and text. This suggests that explicit grounding helps abstract visual reasoning from domain-specific details.

### 4.4.2 Training Steps

Figure 6 tracks the reasoning quality across different reinforcement learning iterations. In the early stages (10 steps), identified points gradually deviated from their correct positions in the chart, leading to reasoning errors. By the final model (100 steps), the correct positions were successfully located, demonstrating concise and evidence-based reasoning chains. Additionally, we observed that more steps led to more diverse but equally correct reasoning steps in textual form. This evolution underscores the critical role of reinforcement learning in extracting effective grounded reasoning strategies from the exploration space.

### 4.4.3 Generalization

Not only can Point-RFT operate effectively in abstract charts, we observe its equal efficacy in real-world scenarios. As illustrated in Figure 7 through two representative examples: In the first case, point-grounded reasoning significantly enhances the model's analytical capabilities when processing complex traffic diagrams, enabling it to recognize fine-grained visual elements and perform logical reasoning. In the second example, the pointing mechanism facilitates explicit expression and reasoning about latent content, allowing accurate interpretation of scenarios that would otherwise be incomprehensible to the original model.

For further analysis on dataset, please refer to the **Supplementary Material**.

## 5 Conclusion

We have presented Point-RFT, a visually grounded Chain-of-Thought framework designed to enhance multimodal document reasoning. Our results demonstrate that structured visual grounding significantly improves accuracy and interpretability in multimodal reasoning, improving the ChartQA accuracy from 70.88% to 90.04%. Among various compared baselines, Point-RFT shows superior generalization capabilities across diverse, out-of-domain benchmarks such as CharXiv, PlotQA, and TabMWP. Additionally, we provide a visually grounded CoT dataset containing 71K samples, facilitating easy format finetuning for future studies on multimodal reasoning. Our findings highlighted the potential on disentanglement of perception errors from reasoning errors, facilitating more precise diagnostics and targeted improvements for large multimodal models.

For statements of broader impact and limitations, please refer to the **Supplementary Material**.

## Acknowledgment

This work was supported by National Key R&D Program of China under Grant No. 2022YFA1004100.

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

## A   Quality of Point-CoT

To verify the quality of Point-CoT, we invited 10 volunteers and GPT-4o itself to validate 1,000 randomly selected samples, calculating the proportion of high-quality samples. We define high-quality samples as those that do not contain reasoning errors and have essentially correct points. As shown in Table 6, we found that the proportion of high-quality samples in Point-CoT approaches 95%, demonstrating the effectiveness of our Point-CoT.

Table 6: **Statistics of dataset.**

| Evaluation | Samples | Ratio |
|---|---|---|
| Human | 1000 | 94.5% |
| Machine | 1000 | 96.8% |

## B   Comparison of Models

Table 7 extends the main paper's Table 2 by comparing two additional baselines, Molmo-7B and Qwen2.5-VL-7B evaluated without the CoT prompting. Superficially, Qwen2.5-VL-7B appears to regress when verbose CoT reasoning is enabled. This drop, however, stems primarily from the model's in-domain short-answer post-training, not from an intrinsic weakness of long-form reasoning—a phenomenon echoed in other RFT studies Deng et al. (2025); Wang et al. (2025a); Huang et al. (2025); Meng et al. (2025); Peng et al. (2025); Chen et al. (2025); Zheng et al. (2025). Nonetheless, our Point-RFT, which couples grounded point-level reasoning with CoT, consistently outperforms both versions of Qwen2.5-VL-7B as well as Molmo-7B. These results confirm that integrating point-based reasoning substantially boosts multimodal reasoning performance, validating the design of both the Point-CoT dataset and the Point-RFT model.

Table 7: **Overall results among different datasets.**

| Method | Setting CoT | In-Domain ChartQA | Out-of-Domain | | | | | |
|---|---|---|---|---|---|---|---|---|
| | | | CharXiv | PlotQA | IconQA | TabMWP | Counting | Avg. |
| Qwen2.5-VL-7B | | 87.30 | 34.40 | **22.30** | 59.10 | 56.40 | 15.00 | 39.06 |
| Qwen2.5-VL-7B | ✓ | 70.88 | 26.50 | 17.80 | 53.40 | 61.00 | 21.00 | 35.94 |
| Molmo-7B | ✓ | 20.48 | 7.60 | 3.95 | 16.50 | 20.70 | 0.00 | 9.75 |
| Point-RFT | ✓ | **90.04** | **36.20** | 20.40 | **59.80** | **70.90** | **78.50** | **53.16** |

## C   Training Steps

As shown in Table 8, SFT requires careful balancing of training steps, reaching peak accuracy at 500 steps (56.72%). Training beyond this point leads to overfitting and performance degradation. For RL, performance monotonically improves with increasing steps, peaking at 100 steps (81.04%). This indicates that extended reward shaping continually refines the reasoning patterns grounded in visual pointing.

## D   Training Dataset

Table 9 reveals that mixing SFT datasets harms performance (81.04% drops to 75.20%). This suggests that targeted format learning on high-quality grounded CoT data is more effective than training on broader but noisier datasets. Notably, Point-RFT even outperforms models trained on pure ChartQA data for both SFT and RL, demonstrating its ability to generalize.

## E   Broader Impact

We present Point-RFT, a multimodal reasoning framework that bridges textual and visual reasoning through visually grounded Chain-of-Thought (CoT). By explicitly anchoring rationales to visual

elements, our approach mitigates hallucinations and enhances multimodal integration, advancing the reliability of AI systems in real-world document analysis. This innovation holds significant potential for human-AI collaboration. Point-RFT paves the way for trustworthy AI assistants capable of seamless multimodal reasoning across diverse domains.

Table 8: **Ablation studies of training steps.**

| Method | Steps | | ChartQA | | |
|---|---|---|---|---|---|
| | SFT | RL | Overall | Inner | Format |
| Base | - | - | 79.28 | 82.17 | 96.48 |
| Point-SFT | 300 | - | 50.00 | 71.88 | 69.56 |
| | 500 | - | **56.72** | **76.20** | **74.44** |
| | 600 | - | 45.56 | 75.48 | 60.36 |
| | 700 | - | 37.00 | 74.71 | 49.52 |
| | 1000 | - | 20.44 | 75.45 | 25.68 |
| Point-RFT | 500 | 5 | 80.92 | 82.14 | 98.52 |
| | 500 | 10 | 83.36 | 83.66 | 99.64 |
| | 500 | 20 | 84.72 | 84.96 | 99.72 |
| | 500 | 50 | 85.48 | 85.58 | **99.88** |
| | 500 | 100 | **86.24** | **86.52** | 99.68 |

# F Limitation

Despite the promising results, our approach has several limitations. First, the two-stage training pipeline (format finetuning and reinforcement finetuning) requires substantial computational resources, particularly when scaling to larger datasets. Moreover, like most autoregressive models, the sequential reasoning process in Point-RFT introduces significant inference latency and resource consumption, especially for complex visual documents. These challenges highlight the need for more efficient training paradigms and inference acceleration techniques. In future work, we will explore lightweight training strategies and parallelizable decoding mechanisms to address these limitations while maintaining reasoning accuracy.

Table 9: **Ablation studies of training dataset.**

| Method | Dataset | | ChartQA | | |
|---|---|---|---|---|---|
| | SFT | RL | Overall | Inner | Format |
| Base | - | - | 79.28 | 82.17 | 96.48 |
| Base-SFT | ChartQA | - | 3.88 | 74.62 | 5.20 |
| Base-RFT | ChartQA | ChartQA | 80.92 | 81.51 | 99.28 |
| Point-RFT | Point-CoT (w/o ChartQA) | ChartQA | 81.04 | 82.42 | 98.32 |
| Point-RFT | Point-CoT | ChartQA | **86.24** | **86.52** | **99.68** |

