# OpenReview forum: "Point-RFT: Improving Multimodal Reasoning with Visually Grounded Reinforcement Finetuning"
_NeurIPS.cc/2025/Conference — NeurIPS 2025 poster_

### Official Review · Reviewer_TioY · 2025-06-29

**Clarity:** 2
**Significance:** 3
**Originality:** 3
**Rating:** 4
**Confidence:** 4

**Summary:**

This paper presents Point-RFT, a two-stage framework for enhancing multimodal large language models (MLLMs) with visually grounded chain-of-thought (CoT) reasoning for visual document understanding. In the first stage, the model is trained to generate visually grounded CoT rationales explicitly aligned with visual elements via point annotations. In the second stage, the model undergoes reinforcement finetuning with Group-wise Relative Policy Optimization (GRPO) to improve answer correctness and generalization. This paper introduces a large, cross-validated Point-CoT dataset and demonstrates substantial gains on ChartQA (in-domain) and several out-of-domain benchmarks, reporting improved accuracy and interpretability over strong baselines.

**Questions:**

Can Point-RFT be applied to or benefit those general image QA understanding tasks, such as Seed-bench2 [1]?

[1] Li, Bohao, et al. "Seed-bench: Benchmarking multimodal large language models." CVPR 2024.

**Ethical Concerns:**

["NO or VERY MINOR ethics concerns only"]

**Final Justification:**

The raised concerns are solved by the authors during rebuttal. Thus, I maintain my positive rating.

**Limitations:**

yes

**Quality:**

3

**Strengths And Weaknesses:**

[Strengths]
1. The idea of extending text-only CoT to visual-grounded CoT is novel and well-motivated.
2. The pipeline for constructing the Point-CoT with cross-model validation mechanism reduces errors caused from unimodal.
3. The analysis of each component in Robot-RFT is comprehensive.

[Weaknesses]
1. **Comparison with Prior SOTA**: The discussion and quantitative comparison with other recent methods for visual document understanding is limited.
2. **Dataset Overlap/Out-of-Domain Clarification**: There is confusion about what is considered out-of-domain. For instance, PlotQA is listed as both a source for training data (Tab. 1) and as out-of-domain in the evaluation tables (Tab. 2 and Tab. 3) .

---

> ### Author Rebuttal · Authors · 2025-07-30
>
> Thank you for your valuable advice and comments!
>
> ## Comparison with Prior SOTA
>
> We have added comparisons with LLaMA Vision 3.2 and Molmo. Specifically, we also trained our Point-RFT using LLaMA Vision 3.2 as the backbone.
>
> |              | Model                | ChartQA  | CharXiv  | PlotQA   | IconQA     | TabMWP   | Counting  | OOD Avg   |
> |--------------|----------------------|----------|----------|----------|------------|----------|-----------|-----------|
> | Base         | Molmo-7B             | 20.5     | 7.6      | 4.0      | 16.5       | 20.7     | 0.0       | 9.8       |
> | Base         | LLaMA-3.2-Vision-11B | 42.1     | 23.1     | 8.5      | 30.0       | 55.0     | 6.0       | 26.5      |
> | Base         | Qwen2.5-VL-7B        | 70.9     | 26.5     | 17.8     | 53.4       | 61.0     | 21.0      | 35.9      |
> | Base-RFT     | Qwen2.5-VL-7B        | 83.9     | 25.2     | 14.9     | 48.6       | 61.4     | 19.0      | 33.8      |
> | Point-SFT    | Qwen2.5-VL-7B        | 87.2     | 28.0     | 20.3     | 55.3       | 60.3     | 70.0      | 46.8      |
> | Point-RFT    | LLaMA-3.2-Vision-11B | **86.2** | **26.5** | **14.4** | **49.0** | **58.4** | **45.0** | **38.7** |
> | Point-RFT    | Qwen2.5-VL-7B        | **90.0** | **36.2** | **20.4** | **59.8** | **70.9** | **78.5** | **53.2** |
>
> The experiments demonstrate the effectiveness of our method across different models. In the revised version, we will add more model comparisons and qualitative discussions.
>
> ## General Image QA Understanding Tasks
>
> We additionally introduced the MMVet dataset as a benchmark, as it is a standardized task suite covering a wide range of multimodal reasoning tasks, further validating our method’s generalization to general images.
>
> |              | Model                | Rec      | Ocr      | Know     | Gen        | Spat     | Math      | Overall   |
> |--------------|----------------------|----------|----------|----------|------------|----------|-----------|-----------|
> | Base         | Qwen2.5-VL-7B        | 34.4     | 49.2     | 14.9     | 12.1       | 52.3     | 59.2      | 40.0      |
> | Point-RFT    | Qwen2.5-VL-7B        | **37.1** | **61.5** | **17.6** | **14.0** | **65.3** | **73.1** | **47.6** |
>
> The results show that our method significantly improves performance across various scenarios, strongly supporting our findings. Although we are unable to include images, we observed that pointing-based thought generalizes natural images well. We will include more examples in the revision.
>
> ## Out-of-Domain Clarification
>
> Sorry for the confusion. Some of the evaluation data categorized as out-of-domain (OOD) also appear in the SFT data. This is because our definition of OOD is based on whether the data is used in the RL phase. Since directly applying reinforcement learning to vision-language models often leads to performance degradation on out-of-domain tasks, using RL data as the criterion allows us to better assess whether our method maintains robust performance beyond the scope of RL-trained data (e.g., ChartQA). We include a broader mix of data types in the SFT stage to train a more generalizable capability, thereby enhancing the model’s usability in real-world scenarios. We will clarify this more explicitly in the revised version.

---

> > ### Comment · Reviewer_TioY · 2025-08-05
> > **Official Comment by Reviewer TioY**
> >
> > Thanks for the detailed rebuttal. My concerns are addressed. I will maintain my positive rating.

---

> > > ### Author Response · Authors · 2025-08-05
> > > **Thank you!**
> > >
> > > We are very appreciated for your thorough review and helping the paper into a better shape!
> > >
> > > Thank you again!
> > >
> > > Authors

---

### Official Review · Reviewer_dht9 · 2025-07-01

**Clarity:** 3
**Significance:** 2
**Originality:** 2
**Rating:** 4
**Confidence:** 4

**Summary:**

This paper proposes Point-RFT, a reinforcement fine-tuning framework that enhances multimodal reasoning by integrating visual grounding into the chain-of-thought (CoT) process. The method proceeds in two stages: first, a supervised format fine-tuning stage using a newly curated dataset (Point-CoT) with step-by-step reasoning grounded to visual elements; second, a reinforcement fine-tuning stage using GRPO to optimize both answer correctness and reasoning format alignment. The paper also introduces a dataset, Point-CoT, a novel multimodal CoT dataset. Experimental results show that Point-RFT significantly improves in-domain accuracy on ChartQA and generalizes well to multiple out-of-domain visual reasoning benchmarks.

**Questions:**

- The paper shows strong generalization on visual document understanding tasks, but it remains unclear whether the proposed Point-RFT framework can extend to broader vision-language reasoning domains. Could the authors evaluate the method on more general VQA-style benchmarks such as MMBench, MMStar, MMVet, or MathVista, which feature diverse visual contexts and question types? A positive result on such benchmarks would strengthen the claim of generalizability and showcase the method's applicability beyond charts and plots.

- In Table 3 (Out-of-Domain), the base Qwen2-VL-7B model achieves a higher average performance than the Qwen2.5-VL series, yet the performance gain after applying Point-RFT appears smaller. Could the authors clarify why the improvement margin differs so much?

- As shown in Table 1, the Point-CoT dataset already includes samples from PlotQA. However, in Table 2 and Table 3, PlotQA is listed as an out-of-domain (OOD) test set. Could the authors clarify this inconsistency?

- The paper includes ablations on format syntax and grounding, but does not isolate the effects of GRPO vs. more standard RL approaches like PPO. Would the benefits of Point-RFT still hold with simpler optimization strategies?

If your response can address my concerns, I will consider increasing my score.

**Ethical Concerns:**

["NO or VERY MINOR ethics concerns only"]

**Final Justification:**

The authors’ rebuttal has largely resolved my concerns, and I will increase my score by one point.

**Limitations:**

yes

**Paper Formatting Concerns:**

no major formatting issue

**Quality:**

3

**Strengths And Weaknesses:**

**Strengths**

The paper focuses on an interesting and meaningful problem. It presents a two-stage training pipeline combining supervised fine-tuning with reinforcement fine-tuning using GRPO. The empirical evaluation is thorough, covering both in-domain (ChartQA) and diverse out-of-domain datasets (e.g., PlotQA, IconQA), with consistent performance gains. The paper is clearly written, well-structured and easy to follow.

**Weakness**

While the use of GRPO in Point-RFT is effective, the paper does not clearly articulate what new insights or innovations it brings beyond prior applications of GRPO in other multimodal or reasoning tasks. A clearer justification of the novelty and insights would strengthen the contribution, and this is my major concern.

The paper shows strong results on visual document understanding benchmarks, but it remains unclear whether Point-RFT can generalize to broader vision-language tasks, such as those involving non-document images. This limitation weakens the overall persuasiveness of the claimed generalization ability of Point-RFT.

Other weaknesses can be found in the Questions section.

---

> ### Author Rebuttal · Authors · 2025-07-30
>
> We appreciate your constructive advice!
>
> ## New Insights or Innovations
>
> **As stated in L61, our core contribution lies in being the first to propose a multimodal language model (VLM) that is outcome reward trained through visually grounded thought.** We demonstrate the advantages of visually grounded chain-of-thought (CoT) reasoning, including:
> - Better scalability compared to pure text-based reasoning in L, shown in Tables 2, 3, and 5;
> - Improved generalization ability, shown in Tables 2 and 3;
> - Better interoperability, shown in Section 4.4.3.
>
> The datasets and engineering implementations are means to achieve these goals, but they are not the core of our aforementioned conceptual contributions.
>
> The core finding in our paper is that, unlike in the text domain, after RL fine-tuning on specific tasks, VLM models exhibit significant in-domain performance gains but often suffer large performance drops on unseen out-of-domain (OOD) tasks, indicating poor generalization. Point-RFT demonstrates that the RL process for VLMs must effectively align reasoning with visual evidence to achieve OOD capabilities. To this end, we train the model to generate “point-and-think” trajectories via SFT and then apply RL, thereby enabling chain-of-thought generalization in RL comparable to LLMs in pure text domains.
>
> As a result, we not only propose an effective dataset and model but also introduce a VLM RL paradigm with cross-task generalization. The real-world examples presented in Section 4.4.3 validate that our method successfully transfers in-domain reasoning ability to real-world domains, effectively addressing the generalization problem of RL in VLMs.
>
> ## Additional Benchmarks on Different Domains
>
> IconQA and PlotQA are related to abstract images and scientific documents instead of standard chart tasks, which demonstrates generalization to document-like content and abstract visual inputs.
>
> We additionally introduced the MMVet dataset as a benchmark, as it is a standardized task suite covering a wide range of multimodal reasoning tasks, further validating our method’s generalization to natural images.
>
> |              | Model                | Rec      | Ocr      | Know     | Gen        | Spat     | Math      | Overall   |
> |--------------|----------------------|----------|----------|----------|------------|----------|-----------|-----------|
> | Base         | Qwen2.5-VL-7B        | 34.4     | 49.2     | 14.9     | 12.1       | 52.3     | 59.2      | 40.0      |
> | Point-RFT    | Qwen2.5-VL-7B        | **37.1** | **61.5** | **17.6** | **14.0** | **65.3** | **73.1** | **47.6** |
>
> The results show that our method significantly improves performance across various scenarios, strongly supporting our findings. Although we are unable to include images, we observed that pointing-based thought generalizes natural images well. We will include more examples in the revision.
>
> ## Clarification on Improvement Margins
>
> The difference in average scores between the two models (42.77 vs. 35.94) is primarily driven by the large performance gap on the counting task (68 vs. 21). The average scores on the other datasets are (36.46 to 38.90) and (39.68 to 46.80), respectively, and show consistent performance.
>
> So, why does the counting task exhibit such an outlier value? This is because Qwen2-VL-7B tends to give direct and correct answers, while Qwen2.5-VL often generates chain-of-thought (CoT) responses that introduce hallucinations. Since the counting task is highly sensitive to hallucinations, Qwen2.5-VL performs worse. However, after applying Point-RFT, Qwen2.5-VL-7B shows reduced hallucinations and its stronger reasoning abilities become more effective, resulting in greater performance gains.
>
> This also demonstrates the advantage of our Point-RFT: through grounded CoT, our method reduces hallucinations and improves reasoning ability during RL.
>
> ## Clarification on OOD Evaluation
>
> Sorry for the confusion. Some of the evaluation data categorized as out-of-domain (OOD) also appear in the SFT data. This is because our definition of OOD is based on whether the data is used in the RL phase. Since directly applying reinforcement learning to vision-language models often leads to performance degradation on out-of-domain tasks, using RL data as the criterion allows us to better assess whether our method maintains robust performance beyond the scope of RL-trained data (e.g., ChartQA). We include a broader mix of data types in the SFT stage to train a more generalizable capability, thereby enhancing the model’s usability in real-world scenarios. We will clarify this more explicitly in the revised version.
>
> ## Evaluation on Other RL Algorithms
>
> We have tried the PPO algorithm on Qwen2.5-VL-7B with Point-RFT.
>
> |              | Model                | RL Algorithm | ChartQA  | CharXiv  | PlotQA   | IconQA     | TabMWP   | Counting  | OOD Avg   |
> |--------------|----------------------|--------------|----------|----------|----------|------------|----------|-----------|-----------|
> | Base         | Qwen2.5-VL-7B        |              | 70.9     | 26.5     | 17.8     | 53.4       | 61.0     | 21.0      | 35.9      |
> | Point-RFT    | Qwen2.5-VL-7B        | PPO          | **88.3** | **34.9** | **19.0** | **56.7** | **65.1** | **74.5** | **50.0** |
> | Point-RFT    | Qwen2.5-VL-7B        | GRPO         | **90.0** | **36.2** | **20.4** | **59.8** | **70.9** | **78.5** | **53.2** |
>
>
> We can find that PPO obtained similar results, showing that the key factor in achieving strong model performance is not the specific RL algorithm itself, but rather our design of the “point-and-think” trajectory and the corresponding outcome reward.

---

> > ### Author Response · Authors · 2025-08-05
> > **Is there any remaining concerns about our paper?**
> >
> > Thank you again for your insightful feedback. It has been invaluable in refining our work. We would be grateful if you could review our rebuttal. Should any questions remain, we would be happy to clarify them.
> >
> > We sincerely appreciate your time and effort.

---

> > ### Comment · Reviewer_dht9 · 2025-08-06
> >
> > Thank you for your detailed response. Your rebuttal has addressed some of my concerns. However, regarding the “Additional Benchmarks on Different Domains,” I still have some doubts. As far as I know, the baseline performance of Qwen2.5-VL-7B on MMVet is significantly higher than 40.0, so the reported result remains unclear to me.
> >
> > In addition, beyond MMVet, some of the other benchmarks mentioned earlier—such as MMStar, MMBench, and MathVista—could provide valuable insights. It would be helpful if the authors could provide more results on these benchmarks to better demonstrate how the proposed method impacts the model’s existing multimodal capabilities.
> >
> > Moreover, in the rebuttal, the authors mention that "Qwen2.5-VL often generates chain-of-thought (CoT) responses that introduce hallucinations. Since the counting task is highly sensitive to hallucinations, Qwen2.5-VL performs worse. " This raises an important question: does this suggest that the CoT generated by Point-RFT only brings positive effects in certain scenarios? It would be helpful if the authors could further clarify this problem.

---

> ### Author Response · Authors · 2025-08-06
>
> Thanks for your comments!
>
> ## Lower Evaluation Scores
>
> This is because, to maintain consistency across all comparisons, we followed the reasoning output setting (i.e., CoT) as described in Lines 202–204 of the main text, which is also the standard setup in prior RFT studies (refer to Section B of **supplementary material**):
>
>  *This drop, however, stems primarily from the model’s in-domain short-answer post-training, not from an intrinsic weakness of long-form reasoning—a phenomenon echoed in other RFT studies (Deng et al., 2025; Wang et al., 2025a; Huang et al., 2025; Meng et al., 2025; Peng et al., 2025; Chen et al., 2025; Zheng et al., 2025).*
>
> We applied the same evaluation standard to all models to ensure a fair comparison (the original score can be found in the next response). Further explanations and comparisons without CoT can be found in Section B of **supplementary material**.
>
> ## MMVet and Other Benchmarks
>
> We choose to report on MMVet because it covers a wider range of question categories and is generally representative of other commonly used benchmarks. Following your suggestion, we have expanded the comparison to include MMVet, MMVista, and MMBench under their original benchmark settings, and found consistent trends across these datasets. Moreover, the results on MMVet particularly highlight our model’s strong ability to transfer reasoning skills from document images to natural images.
>
> | Model            | MMVet  | MMVista | MMBench |
> |------------------|--------|---------|---------|
> | Qwen-VL-2.5-7B   | 67.1   | 68.2    | 82.6    |
> | Point-RFT        | 69.0   | 70.6    | 83.8    |
>
> ## Positive Effect Beyond Specific Scenarios
>
> As demonstrated in Table 2 of the main text, Point-RFT brings substantial improvements across all tasks, not only in counting-related scenarios. Notably, most tasks are unrelated to counting, and the gains are in some cases even more significant on non-counting tasks, further underscoring the generality of our approach. Unless the task is purely textual, multimodal tasks inherently involve perception. In such cases, the CoT generated by Point-RFT consistently improves performance and is not limited to specific scenarios. Our sub-category results on MMVet further support this conclusion.
>
> Looking forward to further discussing with you!

---

> > ### Comment · Reviewer_dht9 · 2025-08-07
> >
> > Thank you for the detailed responses. My concerns have been resolved.

---

> ### Author Response · Authors · 2025-08-07
>
> We are truly delighted to hear that all your concerns have been resolved!
>
> In summary, we have discussed the following original concerns and resolved them as follows:
> - **Novelty**: Clarified that our innovation lies in being the first to combine visually grounded chain-of-thought (CoT) reasoning with reinforcement learning to train VLMs, which significantly enhances cross-task generalization.
> - **Generalization**: Broadened evaluation to MMVet, MMVista, and MMBench, confirming robustness on natural images and varied question types.
> - **Performance Differences**: Showed that Point-RFT dampens CoT-induced hallucinations, with pronounced gains on counting tasks.
> - **OOD Definition**: Defined out-of-distribution as data unseen during the RL phase, ensuring fair generalization tests.
> - **RL Algorithms**: Revealed that improvements stem from the “point-and-think” trajectory design, not the GRPO or PPO.
> - **Task Specificity**: Showed that Point-RFT improves performance across a wide range of tasks and is not limited to any particular scenario.
> - **Benchmark Scores**: Clarified that lower absolute scores were explained by the adoption of unified CoT-based scoring. Detailed comparisons are provided in Suppl. Section B.
>
> We would greatly appreciate it if you could consider raising our score, as all concerns have now been resolved. Should you have any further concerns or questions, we would be more than happy to discuss with you.

---

### Official Review · Reviewer_TsjN · 2025-07-03

**Clarity:** 3
**Significance:** 3
**Originality:** 3
**Rating:** 5
**Confidence:** 4

**Summary:**

This paper proposes Point-RFT, a novel post-training method designed to enhance multimodal reasoning performance. Point-RFT first constructs the Point-CoT dataset using a general-purpose reasoning model, such as GPT-4o, and a grounding expert model, like Molmo-7B. The Point-CoT dataset consists of reasoning trajectories that include coordinates within images and explanations of those regions. Then, a target model is supervised fine-tuned on the constructed dataset. Finally, the SFT-tuned model is further improved through reinforcement learning. The experiments demonstrate that Point-RFT is effective for document understanding tasks, and the results reveal that the point grounding technique is the primary factor behind this effectiveness.

**Questions:**

1. Why does the format training lead to such improvements?
2. How does the proposed method perform when applied to standard VQA tasks? If it shows limited effectiveness or results in performance degradation, what are the underlying reasons?

**Ethical Concerns:**

["NO or VERY MINOR ethics concerns only"]

**Final Justification:**

While I still find this paper somewhat limited in advancing our understanding of the underlying mechanisms of MLLMs, the motivation is sound, and the proposed method demonstrates meaningful improvements. The additional results provided in the response are interesting, though I believe both the number of samples and the range of VQA benchmarks should be expanded to strengthen the findings. Overall, I believe the paper meets the acceptance bar for NeurIPS, and I will keep my score as Accept.

**Limitations:**

yes

**Quality:**

3

**Strengths And Weaknesses:**

**Strrengths**
1. This paper is clear and well-organized overall.
2. The proposed idea is novel and closely mirrors the way humans reason.
3. The experiments demonstrate that the proposed method is effective on in-domain and out-of-domain document understanding tasks.

**Weaknesses**
1. Table 1 shows that Point-SFT already outperforms the Baseline and Base-RFT by a large margin. A deeper analysis of why the format training leads to such improvements would further strengthen the paper’s contribution.
2. How does the proposed method perform when applied to standard VQA tasks? If it shows limited effectiveness or results in performance degradation, what are the underlying reasons?

---

> ### Author Rebuttal · Authors · 2025-07-30
>
> Thank you for your valuable feedback and suggestions!
>
> ## Improvements by Format Training
>
> This is because even in out-of-domain scenarios, using Point-SFT alone can significantly reduce perception and alignment errors, thereby lowering the overall error rate. However, standalone SFT remains constrained by data quality. In contrast, our Point-RFT method based on outcome reward modeling further enhances the quality of visual grounding, leading to improved performance.
>
> To further explore why format training leads to such improvements, we conducted additional experiments where volunteers manually categorized errors on 200 samples that were predicted incorrectly by the base model.
>
> We simply categorized the errors into four types: perception errors, reasoning errors, alignment errors, and other errors. Perception errors represent mistakes in visual perception, such as failing to correctly identify objects or details in an image. Reasoning errors refer to mistakes in logical reasoning or understanding of the text. Alignment errors indicate incorrect referencing of visual information that has been correctly perceived during the reasoning process. Other errors encompass various other types of mistakes.
>
>
> | Error Type               | Base                          | Point-SFT                         |
> |--------------------------|-------------------------------|-----------------------------------|
> | Perception               | 73                            | 29                                |
> | Reasoning                | 61                            | 59                                |
> | Alignment                | 44                            | 13                                |
> | Other                    | 22                            | 20                                |
>
> We found that through format training, the model generates "point-and-think" trajectories during reasoning, significantly improving the alignment between visual information and text reasoning. This leads to a substantial reduction in perception errors and alignment errors, thereby decreasing the overall error rate. This also explains why base-RFT without format training is ineffective.
>
> ## Results on Standard VQA Tasks
>
> We additionally introduced the MMVet dataset as a benchmark, as it is a standardized task suite covering a wide range of multimodal reasoning tasks, further validating our method’s generalization to natural images.
>
> |              | Model                | Rec      | Ocr      | Know     | Gen        | Spat     | Math      | Overall   |
> |--------------|----------------------|----------|----------|----------|------------|----------|-----------|-----------|
> | Base         | Qwen2.5-VL-7B        | 34.4     | 49.2     | 14.9     | 12.1       | 52.3     | 59.2      | 40.0      |
> | Point-RFT    | Qwen2.5-VL-7B        | **37.1** | **61.5** | **17.6** | **14.0** | **65.3** | **73.1** | **47.6** |
>
> The results show that our method significantly improves performance across various scenarios, strongly supporting our findings. Although we are unable to include images, we observed that pointing-based thought generalizes natural images well. We will include more examples in the revision.

---

> > ### Comment · Reviewer_TsjN · 2025-08-03
> >
> > I appreciate the authors' response to my questions. While I still find this paper somewhat limited in advancing our understanding of the underlying mechanisms of MLLMs, the motivation is sound, and the proposed method demonstrates meaningful improvements. The additional results provided in the response are interesting, though I believe both the number of samples and the range of VQA benchmarks should be expanded to strengthen the findings. Overall, I believe the paper meets the acceptance bar for NeurIPS, and I will keep my score as Accept.

---

> > > ### Author Response · Authors · 2025-08-03
> > > **Thank you!**
> > >
> > > We are very appreciated for your thorough review! We will keep improving our paper to make it into a better shape.
> > >
> > > Thank you again!
> > >
> > > Authors

---

### Official Review · Reviewer_EaVN · 2025-07-06

**Clarity:** 3
**Significance:** 3
**Originality:** 2
**Rating:** 3
**Confidence:** 4

**Summary:**

This paper proposes a multimodal reasoning framework that combines with visually
grounded chain-of-thought (CoT) supervision. It includes two-stages: first
constructing a 71K-sample dataset with step-by-step visual grounding, and then
applying GRPO-based RFT to optimize for both reasoning accuracy and format
adherence. Experiments on different datasets demonsrate the effectiveness of the
proposed method.

**Questions:**

See Weaknesses

**Ethical Concerns:**

["NO or VERY MINOR ethics concerns only"]

**Final Justification:**

See official comment.

**Limitations:**

yes

**Quality:**

2

**Strengths And Weaknesses:**

#Strengths

* The studied problem is interesting. The motivation of this paper is clear.

* The empirical performance of the proposed framework is good.

#Weaknesses

* While the proposed framework shows clear engineering effort, the core
contribution is mainly in constructing a new dataset by distilling
GPT-4o-generated reasoning traces with Molmo-7B-based visual grounding. The
novelty here lies more in implementation and data formatting rather than in new
paradigms, and may be considered somewhat incremental.

* The concept of visually grounded chain-of-thought (CoT) is not entirely new —
prior works such as Visual CoT and Sketchpad have explored similar ideas in
different forms.

* The empirical evaluation focuses almost exclusively on chart-based visual
reasoning (e.g., ChartQA, PlotQA, CharXiv), with only a small-scale experiment
on Counting. However, multimodal reasoning is a broad area that spans various
formats such as natural images, science documents, and embodied AI related
tasks. Without testing the method on more diverse benchmarks on different
domains, it’s difficult to evalaute whether the proposed approach generalizes
beyond chart-based data.

* All results are based solely on Qwen-VL models. There is no evaluation
on other model families (e.g., InternVL3 and LLaMA 3.2 vision) or larger models.
It is unclear whether the method scales to stronger models or transfers across
different VLM architectures.

Hu et al., Visual Sketchpad: Sketching as a Visual Chain of Thought for Multimodal Language Models. NeurIPS 2024

Shao et al., Visual CoT: Advancing Multi-Modal Language Models with a Comprehensive Dataset and Benchmark for Chain-of-Thought Reasoning. NeurIPS 2024

---

> ### Author Rebuttal · Authors · 2025-07-30
>
> Thanks for your invaluable advice!
>
> ## Contribution and Novelty Beyond New Dataset
>
> **As stated in L61, our core contribution lies in being the first to propose a multimodal language model (VLM) that is outcome reward trained through visually grounded thought.** We demonstrate the advantages of visually grounded chain-of-thought (CoT) reasoning, including:
> - Better scalability compared to pure text-based reasoning in L, shown in Tables 2, 3, and 5;
> - Improved generalization ability, shown in Tables 2 and 3;
> - Better interoperability, shown in Section 4.4.3.
>
> The datasets and engineering implementations are means to achieve these goals, but they are not the core of our aforementioned conceptual contributions.
>
> The core finding in our paper is that, unlike in the text domain, after RL fine-tuning on specific tasks, VLM models exhibit significant in-domain performance gains but often suffer large performance drops on unseen out-of-domain (OOD) tasks, indicating poor generalization. Point-RFT demonstrates that the RL process for VLMs must effectively align reasoning with visual evidence to achieve OOD capabilities. To this end, we train the model to generate “point-and-think” trajectories via SFT and then apply RL, thereby enabling chain-of-thought generalization in RL comparable to LLMs in pure text domains.
>
> As a result, we not only propose an effective dataset and model but also introduce a VLM RL paradigm with cross-task generalization. The real-world examples presented in Section 4.4.3 validate that our method successfully transfers in-domain reasoning ability to real-world domains, effectively addressing the generalization problem of RL in VLMs.
>
> ## Difference with Visual-CoT and Sketchpad
>
> Our approach relies on **outcome incentivization** rather than **process design or supervision**. This enables broader generalization to diverse tasks, beyond problems with specific handcrafted designs. Specifically, Visual-CoT is limited to one bounding box detected in a narrow domain, and Sketchpad requires external tool, making them difficult to generalize.
>
> ## Diverse Benchmarks on Different Domains
>
> IconQA and PlotQA are related to abstract images and scientific documents instead of standard chart tasks, which demonstrates generalization to document-like content and abstract visual inputs.
>
> We additionally introduced the MMVet dataset as a benchmark, as it is a standardized task suite covering a wide range of multimodal reasoning tasks, further validating our method’s generalization to natural images.
>
>
> |              | Model                | Rec      | Ocr      | Know     | Gen        | Spat     | Math      | Overall   |
> |--------------|----------------------|----------|----------|----------|------------|----------|-----------|-----------|
> | Base         | Qwen2.5-VL-7B        | 34.4     | 49.2     | 14.9     | 12.1       | 52.3     | 59.2      | 40.0      |
> | Point-RFT    | Qwen2.5-VL-7B        | **37.1** | **61.5** | **17.6** | **14.0** | **65.3** | **73.1** | **47.6** |
>
> The results show that our method significantly improves performance across various scenarios, strongly supporting our findings. Although we are unable to include images, we observed that pointing-based thought generalizes natural images well. We will include more examples in the revision.
>
> ## Evaluation on Other Model Families
>
> We choose Qwen2.5-VL as it is a widely used practice for VLM learning. To further validate the effectiveness of our method, we also conducted experiments on LLaMA 3.2 Vision, which also has different model scales.
> |              | Model                | ChartQA  | CharXiv  | PlotQA   | IconQA     | TabMWP   | Counting  | OOD Avg   |
> |--------------|----------------------|----------|----------|----------|------------|----------|-----------|-----------|
> | Base         | LLaMA-3.2-Vision-11B | 42.1     | 23.1     | 8.5      | 30.0       | 55.0     | 6.0       | 26.5      |
> | Point-RFT    | LLaMA-3.2-Vision-11B | **86.2** | **26.5** | **14.4** | **49.0** | **58.4** | **45.0** | **38.7** |
>
> The results demonstrate that our method adapts well to different architectures.

---

> > ### Author Response · Authors · 2025-08-05
> > **Is there any remaining concerns about our paper?**
> >
> > Thank you again for your insightful feedback. It has been invaluable in refining our work. We would be grateful if you could review our rebuttal. Should any questions remain, we would be happy to clarify them.
> >
> > We sincerely appreciate your time and effort.

---

> > ### Comment · Reviewer_EaVN · 2025-08-07
> >
> > I have read the rebuttal and other reviewers' comments. Some of my concerns have been addressed. However, based on the table in the section "MMVet and Other Benchmarks" (from the response to Reviewer dht9), the improvement of the proposed method in the OOD setting under the original benchmark configurations appears quite limited—less than 2%—especially when compared to the substantial gains observed on in-distribution data (over 20%). Therefore, the claim that the proposed method achieves strong OOD generalization might need further studies. Given that OOD generalization is explicitly emphasized in both the paper and the rebuttal, it would be beneficial to include experiments on a wider range of OOD benchmarks (e.g., MMVet, MMVista, and MMBench) using their original evaluation protocols and across different model backbones. Overall, I think this is a borderline submission that requires more rigorous empirical validation. The core idea also appears somewhat incremental, as it builds on a well-established paradigm in LLM research by incorporating incentivized visual grounding into the reasoning trajectory (enhancing visual reasoning by visual grounding is also a well-known method). Accordingly, I am updating my score to 3.

---

### Note · Authors · 2025-08-13

We sincerely thank all reviewers, AC, SAC and PC for their efforts in improving our work.



We appreciate all positive feedback: **clear motivation, novel idea, and interesting problem** (EaVN: "*problem is interesting/motivation … is clear*"; TsjN: "*novel and mirrors human reasoning*"; dht9: "*meaningful problem*"; TioY: "*novel and well-motivated*"), **good performance and generalization** (EaVN: "*empirical performance … is good*"; TsjN: "*meaningful improvements … effective in- and out-of-domain*"; dht9: "*consistent gains … generalizes well*"), **comprehensive analysis** (TioY: "*analysis … is comprehensive*"), **clarity** (TsjN: "*clear and well-organized*"; dht9: "*well-structured and easy to follow*").



We addressed all concerns as follows:

| Reviewer | Concern| Our Response|
|--|--|--|
| EaVN | (i) Novelty; (ii) Eval chart-focused; (iii) Only Qwen-VL tested; (iv) OOD gains only 2% | **(ii) (iii) addressed after discussion.** (i) Clarified that the core contribution is **outcome-incentivized, visually grounded RL** for VLMs; (ii) Added more general benchmarks (MMVet/MMVista/MMBench); (iii) Included LLaMA-3.2-Vision results; (iv) Argued that in general LMM benchmarks, e.g., MMVet, 2–3% gains are **very significant**. |
| TsjN | (i) Why large Point-SFT gains? (ii) Standard VQA performance? | **All addressed after discussion.** (i) Added manual error analysis on 200 base model mistakes; (ii) Added more general benchmarks (MMVet/MMVista/MMBench). |
| dht9 | (i) Insight beyond GRPO; (ii) Need broader VQA; (iii) Backbone gain variance; (iv) OOD/data overlap; (v) RL ablation (PPO vs GRPO). | **All addressed after discussion.** (i) Explained novelty: first to integrate visually grounded CoT with outcome-incentivized RL for cross-task generalization; (ii) Added more general benchmarks (MMVet/MMVista/MMBench); (iii) Explained counting anomaly as CoT hallucination; (iv) Clarified OOD = unseen in RL phase; (v) Added PPO results. |
| TioY | (i) SOTA comparison; (ii) OOD overlap; (iii) General image QA. | **All addressed after discussion.** (i) Added comparisons with Molmo and LLaMA-3.2-Vision; (ii) Clarified OOD definition as above; (iii) Added more general benchmarks (MMVet/MMVista/MMBench) |



Finally, we once again express our sincere gratitude to everyone for your important contributions to improving our work.



Sincerely,

Authors

---

### Decision · Program_Chairs · 2025-09-17

**Decision:**

Accept (poster)

**Comment:**

This paper proposes a post-training enhancement "Point-RFT" for multimodal reasoning and a dataset "Point-CoT" that contains reasoning trajectories with image coordinates and explanations.  The proposed approach is fine-tuned on this dataset.  Experiments show improvements in chart understanding and other visual reasoning benchmarks.

The reviews were leaning positive after the rebuttal and identified strengths of the work such as sound motivation, thorough evaluation (including the results provided during the rebuttal) and analysis.  While major reviewer concerns were addressed, some concerns remain such as expanding the number of samples and the range of VQA benchmarks and OOD VQA benchmarks and different baselines/backbones -- these should be incorporated.